# Vascular Function, Systemic Inflammation, and Coagulation Activation 18 Months after COVID-19 Infection: An Observational Cohort Study

**DOI:** 10.3390/jcm12041413

**Published:** 2023-02-10

**Authors:** Loes H. Willems, Lotte M. C. Jacobs, Laszlo A. Groh, Hugo ten Cate, Henri M. H. Spronk, Boden Wilson-Storey, Gerjon Hannink, Sander M. J. van Kuijk, Chahinda Ghossein-Doha, Magdi Nagy, Dick H. J. Thijssen, André S. van Petersen, Michiel C. Warlé

**Affiliations:** 1Department of Surgery, Radboud University Medical Centre, 6525 GA Nijmegen, The Netherlands; 2Departments of Internal Medicine and Biochemistry, MUMC and CARIM School for Cardiovascular Diseases, 6229 ER Maastricht, The Netherlands; 3Center for Thrombosis and Haemostasis, Gutenberg University Medical Center, 55131 Mainz, Germany; 4Department of Surgery, Bernhoven Hospital, 5406 PT Uden, The Netherlands; 5Department of Medical Imaging, Radboud University Medical Centre, 6525 GA Nijmegen, The Netherlands; 6Department of Clinical Epidemiology and Medical Technology Assessment, Maastricht University Medical Centre, 6202 AZ Maastricht, The Netherlands; 7Department of Obstetrics and Gynaecology, Maastricht University Medical Centre (MUMC), 6229 ER Maastricht, The Netherlands; 8Department of Physiology, Radboud Institute for Health Sciences, Radboud University Medical Centre, 6525 GA Nijmegen, The Netherlands; 9Research Institute for Sport and Exercise Sciences, Liverpool John Moores University, Liverpool L3 5UX, UK

**Keywords:** COVID-19, macrovascular dysfunction, inflammatory cytokines, carotid artery reactivity (CAR)

## Abstract

Introduction: Among its effect on virtually all other organs, COVID-19 affects the cardiovascular system, potentially jeopardizing the cardiovascular health of millions. Previous research has shown no indication of macrovascular dysfunction as reflected by carotid artery reactivity, but has shown sustained microvascular dysfunction, systemic inflammation, and coagulation activation at 3 months after acute COVID-19. The long-term effects of COVID-19 on vascular function remain unknown. Materials and Methods: This cohort study involved 167 patients who participated in the COVAS trial. At 3 months and 18 months after acute COVID-19, macrovascular dysfunction was evaluated by measuring the carotid artery diameter in response to cold pressor testing. Additionally, plasma endothelin-1, von Willebrand factor, Interleukin(IL)-1ra, IL-6, IL-18, and coagulation factor complexes were measured using ELISA techniques. Results: The prevalence of macrovascular dysfunction did not differ between 3 months (14.5%) and 18 months (11.7%) after COVID-19 infection (*p* = 0.585). However, there was a significant decrease in absolute carotid artery diameter change, 3.5% ± 4.7 vs. 2.7% ± 2.5, *p*—0.001, respectively. Additionally, levels of vWF:Ag were persistently high in 80% of COVID-19 survivors, reflecting endothelial cell damage and possibly attenuated endothelial function. Furthermore, while levels of the inflammatory cytokines interleukin(IL)-1RA and IL-18 were normalized and evidence of contact pathway activation was no longer present, the concentrations of IL-6 and thrombin:antithrombin complexes were further increased at 18 months versus 3 months (2.5 pg/mL ± 2.6 vs. 4.0 pg/mL ± 4.6, *p* = 0.006 and 4.9 μg/L ± 4.4 vs. 18.2 μg/L ± 11.4, *p* < 0.001, respectively). Discussion: This study shows that 18 months after COVID-19 infection, the incidence of macrovascular dysfunction as defined by a constrictive response during carotid artery reactivity testing is not increased. Nonetheless, plasma biomarkers indicate sustained endothelial cell activation (vWF), systemic inflammation (IL-6), and extrinsic/common pathway coagulation activation (FVII:AT, TAT) 18 months after COVID-19 infection.

## 1. Introduction

A substantial amount of research has been performed to uncover the mechanism of action and the consequences of the severe acute respiratory syndrome coronavirus 2 (SARS-CoV-2). COVID-19 mostly affects the upper airways and pulmonary system [1]. However, virtually all organs can be affected. Among others, COVID-19 induces inflammation, platelet activation, hypercoagulability, and endothelial cell (EC) dysfunction, which contribute to the occurrence of complications [2,3]. As a result, COVID-19 may jeopardize the cardiovascular health of millions [4].

Previous results from our group revealed that at 3 months after acute COVID-19, sustained EC involvement, inflammation, and coagulation activity were present [5]. A heightened innate immune response state and a prothrombotic state promote macrovascular EC dysfunction and damage, which impair important functions of the endothelium [6,7,8], resulting in an increased risk of developing major arterial thrombotic events such as myocardial infarction or stroke [9,10]. The literature already revealed a higher incidence of large-vessel stroke [11,12] and myocardial infarction [13] in acute COVID-19 patients.

Our research group found no indication of macrovascular dysfunction at 3 months after acute COVID-19; however, macrovascular dysfunction might take a longer time to develop [5]. The effect of COVID-19 infection on ECs has been studied elaborately, but the long-term effects of the disease on ECs and blood vasculature remain to be determined [7,14,15]. Therefore, this study aimed to determine the long-term effects (18 months) of acute COVID-19 infection on macrovascular endothelial function. Additionally, this study reveals whether EC involvement, elevated circulating inflammatory cytokines, and coagulation system activation remain present in patients 18 months after recovery.

## 2. Materials and Methods

### 2.1. Study Design and Participants

The COVAS study was an observational cohort study, initiated at Radboudumc (Nijmegen, The Netherlands) and conducted at Bernhoven hospital (Uden, The Netherlands). Detailed description of the design, inclusion and exclusion criteria, methods, and cross-sectional data at 3 months post-COVID have been published previously [5]. In brief, patients aged 16 years or older who had experienced SARS-CoV-2 infection were recruited. Patients with a recent (<3 months) episode of angina pectoris, myocardial infarction, stroke or heart failure, or upper-extremity conditions contra-indicating cold pressor testing were excluded. The study was approved by the Medical Regional Ethics Committee Oost-Nederland (reference number: NL74101.091.20), and local approval was obtained from the local directory boards. This study was conducted in accordance with the latest revision of the Declaration of Helsinki. Written informed consent was obtained from all participants before the start of any study-related procedures. Data are available upon request from the authors.

### 2.2. Procedures and Endpoints

Participants of the COVAS study were followed for a period of 18 months after recovery from acute COVID-19 symptoms. The first visit was planned at least 6 and no more than 20 weeks after recovery from acute COVID-19. The second study visit was planned at 18 months after COVID-19 infection. During both visits, macrovascular endothelial function was assessed using the carotid artery reactivity (CAR) test and whole blood was collected for determining markers of microvascular EC dysfunction, inflammation, and coagulation system activation.

### 2.3. Measurement of Macrovascular Dysfunction via the Carotid Artery Reactivity (CAR) Test

The CAR test is a non-invasive procedure, measuring the response of the carotid artery diameter to cold pressor testing (CPT), a 3 min immersion of the hand in ice water (≤4.0 °C) [5,16,17,18]. Carotid artery reactivity was classified as dilatation (normal endothelial function) or constriction (macrovascular endothelial dysfunction) and the relative carotid artery diameter change from baseline (CAR%) was determined to quantify this response. Recorded videos of the carotid artery diameter were analyzed using edge detection and wall tracking software, custom-designed by Philips Healthcare (Best, The Netherlands). Details on how this test was performed can be found elsewhere [5]. In short, the left common carotid artery was visualized using the L12-4 MHz linear array probe of the Philips Lumify ultrasound device with the participant lying in the supine position. Participants were asked to adhere to several guidelines prior to CAR testing: (1) no food or drinks other than water within 6 h prior to the study visit; (2) no coffee, tea, soft drinks, alcohol, vitamin-C-rich products, or chocolate within 18 h prior to the study visit; and (3) no intensive physical exercise within 24 h prior to the study visit.

### 2.4. Determining Markers of Microvascular EC Dysfunction, Inflammation, and Coagulation System Activation

Platelet poor plasma was obtained from whole blood samples after centrifuging for 30 min in two steps (first 10 min, second 20 min), both of 2500× *g* at room temperature. The plasma was snap-frozen at −80 °C until further assay. Concentrations of endothelin-1 (ET-1), interleukin-1 receptor antagonist (IL-1RA), IL-6, and IL-18 were measured according to the manufacturer’s protocol using commercial quantikine ELISA kits (R&D Systems, Minneapolis, MN, USA). Concentrations of von Willebrand factor antigen (VWF:Ag) and activated coagulation factors in complex with their natural inhibitors were measured using in-house developed ELISA methods [5,19]. Measured coagulation factor inhibitor complexes included thrombin:antithrombin (TAT), factor(F)VIIa:AT, FIXa:AT, FXIa:AT, FXIa:alpha-1-antitrypsin (α1AT) and FXIa:C1-esterase-inhibitor (C1inh).

### 2.5. Adverse Cardiovascular Events

During the second study visit at 18 months after COVID-19 infection, participants were asked if they experienced any adverse events after their first study visit. Specifically, participants were asked if they had experienced myocardial infarction, stroke, acute limb ischemia, revascularization procedures, or above-ankle amputation of a lower extremity. Additionally, patient files of all participants were checked for the before-mentioned events and death.

### 2.6. Statistical Analysis

The change in the proportion of participants with vasoconstriction during the CAR test at 3 months versus 18 months was tested using McNemar’s test. Changes in CAR%, markers of microvascular EC dysfunction, markers of inflammation, and markers of coagulation system activation at 3 months versus 18 months were compared using the Wilcoxon signed rank test. The proportion of participants with concentrations of previously mentioned markers above the normal range was reported. Normal ranges of in-house developed ELISA methods were defined as above-normal mean ± 1 SD, based on previous validation studies [5,19]. Analyses were performed using IBM SPSS Statistics 27. *p*-values below 0.05 were considered to be statistically significant.

## 3. Results

### 3.1. Study Population

During the first study visit, 203 patients provided written informed consent. All participants were invited for a follow-up visit 18 months after acute COVID-19 and 167 participants accepted the invitation and visited Bernhoven hospital between 1 November 2021 and 16 February 2022. Thirty-six participants refused the invitation and did not complete the study due to loss of follow-up (4), withdrawal of consent (16), researcher’s decision (2), death (4), or other reasons (10). Figure 1 demonstrates the study flowchart. Baseline patient characteristics are shown in Table 1.

### 3.2. No Indication of Delayed Onset Macrovascular Dysfunction, but Reduced CAR% at 18 Months after COVID-19

No differences in macrovascular dysfunction as assessed by CAR testing were found when comparing the incidence at 3 months to 18 months after acute COVID-19 (Table 2). At 3 months after acute COVID-19, vasoconstriction occurred in 14.5% of participants and at 18 months after acute COVID-19 in 11.7% of participants, *p* = 0.584 (Table 2). There was a significant decrease, however, in CAR% between 3 months and 18 months after infection, 3.5 ± 4.7% versus 2.7 ± 2.5%, respectively.

### 3.3. Partial Normalization of Microvascular EC Dysfunction, Inflammation, and Contact Pathway Activation, but Further Increase in IL-6 and Extrinsic Pathway Activation at 18 Months versus 3 Months after COVID-19 Recovery

ET-1 as a marker of microvascular EC dysfunction showed a significant decrease in plasma concentrations with mean levels being in the normal range, while vWF:AG levels, representing endothelial damage, remained elevated in the majority of patients at 18 months after acute COVID-19. IL-1RA and IL-18, which were both highly activated at 3 months after COVID-19, were significantly decreased and restored to normal means at 18 months after COVID-19. However, IL-6 appeared to be further elevated at 18 months versus 3 months after acute COVID-19, 4.0 ± 4.6 pg/mL vs. 2.5 ± 2.6 pg/mL (*p* = 0.006), with 19.4% of participants having IL-6 levels above the normal range at 18 months after COVID-19. Predominantly, a normalization of markers of contact activation was demonstrated with a decrease in the mean concentrations of FIXa:AT, FXIa:α1AT, and FXIa:C1inh below the upper limit of normal and a decrease, but not normalization, of FXIa:AT. On the other hand, TAT, reflecting ongoing thrombin generation, as well as FVIIa:AT, an extrinsic pathway activity marker, were further elevated: 4.9 ± 4.4 vs. 18.2 ± 11.4 (*p* < 0.001) and 411.9 ± 317.0 vs. 426.8 ± 356.0 (*p* = 0.005), respectively (Table 3).

### 3.4. Major Adverse Cardiovascular Events

Some participants experienced adverse cardiovascular events during the time between the study visits. Myocardial infarction occurred in one participant and one participant had a stroke. Two participants underwent revascularization. There was no significant difference between participants that did or did not experience a major adverse cardiovascular event for any of the variables.

## 4. Discussion

This study revealed that COVID-19 does not cause delayed onset macrovascular endothelial dysfunction, as measured using carotid artery reactivity testing. Nonetheless, there is still evidence of sustained endothelial cell damage with largely restored levels of ET-1, but persistent high levels of vWF:Ag. Sustained inflammation and coagulation activation as previously demonstrated at 3 months after acute COVID-19 were partly restored in the long-term (18 months). However, further increases in IL-6 levels and TAT were present.

CAR testing revealed no increased macrovascular dysfunction reflected by constriction of the carotid artery as a reaction to CPT. However, the CAR% was significantly reduced at 18 months compared to 3 months after acute COVID-19. As the vascular endothelium is critical for maintaining vascular tone and homeostasis, endothelial dysfunction switches the vascular equilibrium to vasoconstriction [14,15]. The decrease in CAR% could therefore represent COVID-19-induced reduced vascular wall relaxation capacity. The clinical importance of such changes in CAR%, without influencing the prevalence of macrovascular dysfunction, may not be immediately evident. Additionally, other factors might have influenced the CAR%. Previous research has described a decreased vasomotor response with advancing age [17,21]. In the current study, the age difference between follow-up moments was 15 months, which might have slightly influenced the CAR%. Another explanation might be the difference in adherence to the guidelines prior to CAR testing. At 3 months, 75% of participants adhered to all guidelines, while at the second study visit only 57% of participants adhered to all guidelines. The guideline that was adhered to the least was not drinking coffee, tea, or soft drinks 18 h prior to the CAR test, with 17% and 32% of participants not following these instructions at 3 months and at 18 months after COVID-19 infection, respectively. Coffee exerts an acute unfavorable effect on endothelial function [22] that might have resulted in a decrease in CAR%.

Mean blood plasma levels of inflammatory cytokines IL-1RA and IL-18 were lower and no longer elevated at 18 months versus 3 months after COVID-19 infection. This suggests at least a partial restoration of the chronic low-grade inflammation state that was present 3 months after COVID-19 infection [5]. Circulating IL-18 is moderately and independently associated with cardiovascular disease (CVD) and, also, higher levels of IL-1RA are positively associated with incident CVD [23,24]. Therefore, a reduction in plasma concentrations of IL-18 and IL-1RA may be beneficial for the cardiovascular health of COVID-19 survivors.

Most notably, however, data from this study also indicate a significant increase in blood plasma levels of the inflammatory cytokine IL-6 between 3 months and 18 months after COVID-19 recovery. The amount of participants with IL-6 levels above the normal range compared to the literature increases from 3.7% to 19.4%. Compared to their own levels, the plasma concentrations increase by 60% in the 15 months following the first study visit, 2.5 ± 2.6 pg/mL versus 4.0 ± 4.6 pg/mL, *p* = 0.006. COVID-19 infection can cause endothelial dysfunction and thrombosis by two proposed mechanisms: acting directly on the endothelium and impairing its anti-thrombogenic and barrier properties, or acting indirectly through a local cytokine storm and systemic inflammation resulting in endothelial injury [7]. Cytokines are believed to play a major role during the control and resolution of COVID-19 infection [25]. Most infected patients develop mild to moderate symptoms, but in some COVID-19 patients, exacerbated pro-inflammatory cytokine release occurs, known as cytokine release syndrome (CRS). CRS results in systemic inflammation and can in turn lead to multiple organ failure [25,26,27]. IL-6 plays an important role in CRS and is positively correlated with COVID-19 severity [26,28]. Persisting and even further elevation of IL-6 levels in the long-term post-COVID might be an alarming signal of a persistent COVID-19-induced inflammatory state. An important sidenote is that the COVID-19 booster vaccination campaign was initiated during the course of this study. Vaccination against COVID-19 has been shown to increase IL-6 levels with an early small peak 1 day after vaccination [29,30]. Recent research demonstrates that a vaccination-induced IL-6 level increase is no longer present at 4 weeks after vaccination [31]. The exact number of patients that were vaccinated in the days or weeks preceding the second study visit is unknown. The influence of IL-6 levels in study participants is expected to be limited since the peak is described to be short and early. Still, the COVID-19 vaccination may have resulted in higher levels of IL-6 at 18 months versus 3 months post-COVID-19 in some participants. No changes in IL-18 levels have been observed after COVID-19 vaccination [29]. Another important issue is that ageing is associated with a chronic progressive increase in the proinflammatory status, called inflammaging [32]. There is strong evidence that serum concentrations of the pro-inflammatory cytokine IL-6 increase with age [33]. However, since the age difference between follow-up moments was only 15 months, and the increase was not seen in other inflammatory cytokines, the effect of inflammaging is expected to be limited in our cohort.

Ongoing contact activation has been confirmed in patients with active COVID-19 disease with a clear link to disease severity, possibly explaining the high incidence of thrombotic complications in severe COVID-19 [34]. After 3 months, only a minority of participants showed elevated markers of contact activation, where long-term follow-up demonstrated further normalization. On the other hand, TAT, a marker of thrombin generation, is further increased at 18 months compared to 3 months after acute COVID-19. Other studies have demonstrated evidence of increased thrombin generation up to 1 year after acute COVID-19 infection [35,36], proposedly caused by the underlying mechanisms of persistent endothelial damage. Indeed, in the current study, we found evidence of the sustained elevation of vWF:Ag levels, with 80% of participants still having vWF:Ag levels above the normal range. Additionally, we found increased levels of IL-6 which is a driver of tissue factor expression. Tissue factor is an initiator of the extrinsic coagulation pathway, ultimately leading to thrombin generation [37]. We found increased levels of markers of both extrinsic pathway activity (FVII:AT) and thrombin generation (TAT).

Since previous research on COVID vaccination demonstrates no effect on TAT, the booster vaccination campaign initiated during this study is an unlikely explanation for the ongoing elevation of TAT.

This study has some limitations. First, adherence to guidelines prior to CAR testing at the second study visit was somewhat modest compared to the first study visit. Participants were asked to adhere to several guidelines prior to CAR testing to minimize the influence of oral intake and behavior on endothelial function. At 3 months, 75% of participants adhered to all guidelines, compared to 57% at 18 months. The difference was mostly explained by the intake of coffee, tea, or (possibly caffeine-containing) soft drinks in the 18 h preceding CAR testing. This might have led to a slight deviation from the true effect. However, this deviation would be to the disadvantage of the second follow-up visit, where a lower prevalence of macrovascular dysfunction was observed. Therefore, we are convinced that there is no indication of late-onset macrovascular dysfunction at 18 months after COVID-19 infection. Second, information about factors which could influence IL-6 levels in the blood, such as recent vaccinations or re-infection with COVID-19, were not collected from the participants. COVID-19 vaccination leads to an increase in IL-6 for at least 1 day, but no longer than 4 weeks [29,30,31]. The effect of re-infection on IL-6 concentrations is not well described in the literature, but in primary infection, a rise in cytokines appears to play a major role [25]. Since the booster campaign was initiated during the study period, and the pandemic continued, we cannot be sure whether or not the observed increase in IL-6 concentrations was caused by COVID-19 vaccination or re-infection.

In conclusion, this study showed that COVID-19 does not increase the incidence of macrovascular dysfunction as assessed by carotid artery reactivity testing until 18 months after infection. Nevertheless, our results indicate that, 18 months after COVID-19, there is evidence of (1) sustained EC damage with persistent high levels of vWF:Ag in almost all survivors and a slight increase in CAR% which might indicate attenuated endothelial function; (2) a sustained and further rise in circulating IL-6 levels compared to 3 months after COVID-19, possibly indicating a chronic low-grade inflammatory state; and (3) no longer evidence of contact pathway activation but a further rise in the marker of thrombin generation TAT and the extrinsic pathway marker FVII:AT, which indicates a persistent prothrombotic state, possibly triggered by increased tissue factor expression through IL-6. Therefore, based on the findings of the current study, the long-term cardiovascular risk after acute COVID-19 infection could be increased. Future research should focus on the clinical consequences of the chronic inflammatory and prothrombotic state, such as cardiovascular complications.

## Figures and Tables

**Figure 1 jcm-12-01413-f001:**
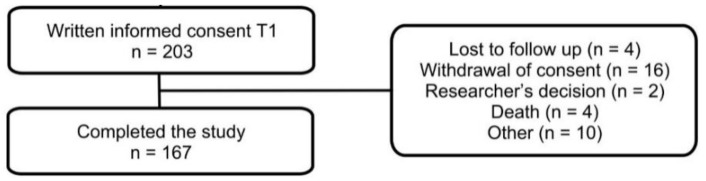
Flowchart of the study population. T1: 3 months after acute COVID-19.

**Table 1 jcm-12-01413-t001:** Baseline characteristics of all participants who completed the study.

		*n* = 167
Patient characteristics	Age, mean ± SD	62.3 ± 12.1
	Male sex, *n* (%)	112 (76.1)
	BMI, mean ± SD	27.8 ± 4.1
	History of smoking, *n* (%)	112 (67.1)
Comorbidities	Hypertension, *n* (%)	71 (42.5)
	Hyperlipidemia, *n* (%)	45 (26.9)
	Cardiovascular disease, *n* (%)	68 (40.7)
	Diabetes mellitus, *n* (%)	20 (12.0)
Disease severity	Days of illness, median [range]	19 [1–80]
	Hospital care, *n* (%)	109 (65.3)
	Days, median [range]	6 [1–61]
	Intensive care, *n* (%)	24 (14.4)
	Days, median [range]	16 [1–42]

SD = standard deviation, BMI = body mass index.

**Table 2 jcm-12-01413-t002:** Total participants with a vasodilatation and vasoconstriction response at 3 months and 18 months after acute COVID-19.

		CAR Response at T2, 18 Months after Acute COVID-19		
		Dilatation	Constriction	Total
CAR response at T1, 3 months after acute COVID-19	Dilatation	111	13	124
	Constriction	17	4	21
	Total	128	17	145
Constriction, n (%)				
3 months after acute COVID-19		21 (14.5)		*p* = 0.584 †
18 months after acute COVID-19		17 (11.7)		
CAR%, mean ± SD				
3 months after acute COVID-19		3.5 ± 4.7		*p* = 0.001 ‡
18 months after acute COVID-19		2.7 ± 2.5		

CAR = carotid artery reactivity, SD = standard deviation. † The change in the proportion of participants with vasoconstriction during the CAR test was tested using McNemar’s test. ‡ Changes in CAR% were compared using the Wilcoxon signed rank test.

**Table 3 jcm-12-01413-t003:** Endothelial cell activation, circulating inflammatory cytokines, and markers of coagulation system activation at 3 months versus 18 months after recovery of acute COVID-19.

	Normal Range	T1 (+3 Months)		T2 (+18 Months)		*p*
		Mean ± SD	High, %	Mean ± SD	High, %	
ET-1	0.87–1.61 pg/mL	2.56 ± 1.42	68.3	1.44 ± 0.53	30.5	<0.001
VWF:Ag	≤160%	273.2 ± 138.3	81.0	244.2 ± 89.6	80.0	0.090
IL-1RA	100–400 pg/mL	494.8 ± 383.0	47.9	363.5 ± 541.4	22.5	<0.001
IL-6	4.6–5.7 pg/mL [20]	2.5 ± 2.6	3.7	4.0 ± 4.6	19.4	0.006
IL-18	37–215 pg/mL	307.0 ± 126.9	75.4	207.8 ± 126.5	34.5	<0.001
TAT	≤4.0 μg/L	4.9 ± 4.4	47.2	18.2 ± 11.4	95.7	<0.001
FVIIa:AT	237.7–374.6 pg/mL	411.9 ± 317.0	34.4	426.8 ± 356.0	35.0	0.005
FIXa:AT	187.3–265.9 pg/mL	255.2 ± 66.0	27.6	234.6 ± 62.5	25.2	<0.001
FXIa:AT,	7.0–12.5 pg/mL	22.7 ± 85.2	16.0	15.3 ± 28.8	4.9	<0.001
FXIa:α1AT	78.6–120.1 pg/mL	163.4 ± 487.5	19.0	70.6 ± 6.8	0.6	<0.001
FXIa:C1inh	176.7–396.7 pg/mL	500.2 ± 1604.3	17.8	348.7 ± 814.0	11.7	<0.001

ET-1 = endothelin-1; VWF:Ag = von Willebrand factor antigen; IL-1RA = interleukin 1 receptor antagonist; TAT = thrombin:antithrombin; F = factor; α1AT = alpha-1-antitrypsin; C1inh = C1-esterase-inhibitor.

## Data Availability

The raw data supporting the conclusions of this article will be made available by the authors upon reasonable request.

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
