# Peer review of "Vascular Function, Systemic Inflammation, and Coagulation Activation 18 Months after COVID-19 Infection: An Observational Cohort Study"

_jcm, 2023, doi:10.3390/jcm12041413_

Round 1

Reviewer 1 Report

This is an interesting and well written paper that fits the Journal scope. The authors analyze the vascular function, systemic inflammation, and coagulation activation 18 months after COVID-19 infection.  

This study shows that 18 months after COVID-19 infection, the incidence of macrovascular dysfunction as defined by a constrictive response during carotid artery reactivity testing is not increased. Nonetheless, plasma biomarkers indicate sustained endothelial cell activation (vWF), systemic inflammation (IL-6) and extrinsic/common pathway coagulation activation (FVII:AT, TAT) 18 months after COVID-19 infection.

Well Done.

Author Response

We thank the reviewer for his/her time to review our paper and positive judgement.

Reviewer 2 Report

The bibliographical references are not homogeneous, since some include the month and others do not, as well as the country.

It is important to take into consideration that, despise the fact that some inclusion and exclusion criteria are controlled in the study, in the CAR testing, the study does not determine whether the changes observed in the results, as well as adverse cardiovascular events are related to the characteristic of the patients, such as comorbidities; since a high  percentage has history of smoking, hypertension, hyperlipidemia, cardiovascular disease and diabetes mellitus, or serious illness and required time and hospital care.

On the other hand, the question remains whether the IL-6 concentration at 18 months 4.0+/-4.6 compared to the normal range 4.65+/-5.7 pg/mL can be considered above the normal range.

Author Response

The bibliographical references are not homogeneous, since some include the month and others do not, as well as the country.

Response: we thank the reviewer for his/her critical view. We now mention the month in all references. Country/city is mentioned in a few references, but only if it’s part of the name of the journal

It is important to take into consideration that, despite the fact that some inclusion and exclusion criteria are controlled in the study, in the CAR testing, the study does not determine whether the changes observed in the results, as well as adverse cardiovascular events are related to the characteristic of the patients, such as comorbidities; since a high  percentage has history of smoking, hypertension, hyperlipidemia, cardiovascular disease and diabetes mellitus, or serious illness and required time and hospital care.

Response: we agree with the reviewer that elevated levels of biomarkers are not necessarily related to COVID-19, but may be related to characteristics of patients. However, significant changes over time are most likely to be related to other influences, since the characteristics of patients do not vary over time, except for age. We described the influence of age on our primary outcome measurements in the discussion section [page 12, line 20-22]. Also levels of inflammatory cytokines can be influenced by age. We added a section on this phenomena in our discussion,  page 14, lines 13-18.

“Another important issue is that ageing is associated with a chronic progressive increase in the proinflammatory status, called inflammaging.32 There is strong evidence that serum concentrations of the pro-inflammatory cytokine IL-6 increase with age.33 However, since the age difference between follow-up moments was only 15 months, and the increase was not seen in other inflammatory cytokines, the effect of inflammaging is expected to be limited in our cohort.”

Adverse cardiovascular events were only experienced by 4 of our participants. We cannot prove nor intend to claim that these events have a causal relation with COVID-19.

On the other hand, the question remains whether the IL-6 concentration at 18 months 4.0+/-4.6 compared to the normal range 4.65+/-5.7 pg/mL can be considered above the normal range.

Response: this is a valid point of the reviewer. Approximately 20% of participants have IL-6 levels above normal range, leaving 80% of participants with IL-6 levels not above normal range compared to means in literature. Compared to their own levels 15 months earlier, however, we observe an increase of 60%. We clarified the findings concerning IL-6 in our discussion section, page 13, lines 15-18.

“The amount of participants with IL-6 levels above normal range compared to literature increases from 3.7% to 19.4%. Compared to their own levels, the plasma concentrations increase with 60% in the 15 months following first study visit, 2.5 ± 2.6 pg/mL versus 4.0 ± 4.6 pg/mL, p=0.006.”

Reviewer 3 Report

The Paper is interesting because presents macrovascular  dysfunction by carotid reactivity and plasma biomarkers of endotelial cell dysfunction  inflammation and coagulation system at 18 months after Covid 19.The Parer is ell written .The distribution of introduction,methods, discussion are well presentd  .

Line 101  in methods  How the CAR test was performed and  the probe used  must be reported.

The discussion on limitations as low adherence to guidelines and information about vaccination or reinfection merit better discussion.

Table 2 is not clear,Some references do not follow regulation of journal as  referencee 33 and 34

Author Response

Response: we thank the reviewer for the time and the compliments.

Line 101  in methods  How the CAR test was performed and  the probe used  must be reported.

Response: we added information on how the CAR test was performed on page 7, lines 7-10

Details on how this test was performed can be found elsewhere.5 In short, the left common carotid artery was visualized using L12-4 MHz linear array probe of Philips Lumify ultrasound device with the participant laying in supine position.”

The discussion on limitations as low adherence to guidelines and information about vaccination or reinfection merit better discussion.

Response: we dedicated additional lines to this section of the discussion, page 15, lines 12-17 and 24, page 16 lines 1-3

“This study has some limitations. First, adherence to guidelines prior to CAR testing at the second study visit was somewhat modest compared to the first study visit. Participants were asked to adhere to several guidelines prior to CAR testing to minimize the influence of oral intake and behaviour on endothelial function. At 3 months, 75% of participants adhered to all guidelines, compared to 57% at 18 months. The difference was mostly explained by the intake of coffee, tea, or (possibly caffein containing) soft drinks in the 18 hours preceding CAR testing.”

“COVID-19 vaccination leads to an increase in IL-6 for at least 1 day, but no longer than 4 weeks.29-31 The effect of re-infection on IL-6 concentrations is not well described in literature, but in primary infection, rise in cytokines appear to play a major role.25 Since the booster campaign was initiated during the study period, and the pandemic continued, we cannot be sure whether or not the observed increase in IL-6 concentrations were caused by COVID-19 vaccination or re-infection.”

Table 2 is not clear

Response: we have tried to clarify table 2 by removing unnecessary information and highlighting some important separation lines

Some references do not follow regulation of journal as  referencee 33 and 34

Response: we thank the reviewer for his/her critical view. We revised the references.